**Data Availability Statement:** All relevant data are within the paper and its Supporting Information files.

# Soluble chitosan derivative treats wound infections and promotes wound healing in a novel MRSA-infected porcine partial-thickness burn wound model

**Francesco Egro**[1]◉, **Alex Repko**[1], **Vidya Narayanaswamy**[2]◉, **Asim Ejaz**[1], **Deokyeol Kim**[1], **M. Asher Schusterman**[1], **Allister Loughran**[2¤a], **Ali Ayyash**[1], **Stacy M. Towsend**[2¤b], **Shenda Baker**[2], **Jenny Ziembicki**[3], **Kacey Marra**[1,4,5], **Peter Rubin**[1,4,5]*

1 Department of Plastic Surgery, University of Pittsburgh, Pittsburgh, PA, United States of America, 2 Synedgen Inc., Claremont, CA, United States of America, 3 Department of Surgery, University of Pittsburgh Medical Center Mercy, Pittsburgh, PA, United States of America, 4 Department of Bioengineering, University of Pittsburgh, Pittsburgh, PA, United States of America, 5 McGowan Institute of Regenerative Medicine, Pittsburgh, PA, United States of America

◉ These authors contributed equally to this work.
¤a Current address: St. Jude Children's Research Hospital, Memphis, TN, United States of America
¤b Current address: Townsend Biopharm Consulting, Burlingame, California, United States of America
* drrubinp@gmail.com

## Abstract

Burns are physically debilitating and potentially fatal injuries. The most common etiology of burn wound infections in the US is methicillin-resistant *Staphylococcus aureus* (MRSA), which is particularly recalcitrant when biofilms form. The current standard of care, silver sulfadiazine (SSD) is effective in reducing bacterial load, but less effective in improving burn wound healing. New treatments that can manage infection while simultaneously improving healing would provide a benefit in the treatment of burns. Porcine models are frequently used as a model for human wound healing but can be expensive due to the need to separate wounds to avoid cross contamination. The porcine model developed in this study offers the capability to study multiple partial thickness burn wound (PTBW) sites on a single animal with minimal crosstalk to study wound healing, infection, and inflammation. The current study evaluates a wound rinse and a wound gel formulated with a non-toxic, polycationic chitosan derivative that is hypothesized to manage infection while also promoting healing, providing a potential alternate to SSD. Studies *in vitro* and in this PTBW porcine model compare treatment with the chitosan derivative formulations to SSD. The wound rinse and wound gel are observed to disrupt mature MRSA biofilms *in vitro* and reduce the MRSA load *in vivo* when compared to that of the standard of care. *In vivo* data further show increased re-epithelialization and faster healing in burns treated with wound rinse/gel as compared to SSD. Taken together, the data demonstrate the potential of the wound rinse/gel to significantly enhance healing, promote re-epithelialization, and reduce bacterial burden in infected PTBW using an economical porcine model.

**Funding:** The work was funded by the United States Department of Defense, through the Armed Forces Institute of Regenerative Medicine (AFIRM), under Award No. W81XWH-14-2-0004, awarded to PR. Synedgen's funding for SynePure Wound Cleanser was provided by DARPA (grant numbers N66001-14-C-4009 and N66001-12-C-4053) awarded to SB, and funding for the Catasyn Advanced Technology Wound Hydrogel was provided by the US Army (grant number W81XWH-13-C-0053 and W81XWH-16-C-0023) awarded to SB. The funders had no role in study design, data collection and analysis, decision to publish, or preparation of the manuscript.

**Competing interests:** VPN and SMB are paid employees of Synedgen. SMB have ownership and patents affiliated with Synedgen and is also a board member. The potential conflicts noted have not impacted or influenced the findings of this manuscript. For the remaining authors none are declared. This does not alter our adherence to PLOS ONE policies on sharing data and materials.

## Introduction

Burn injury is among the most common injuries reported worldwide. In the United States, 500,000 cases of burns require medical care every year, of which, $\sim$ 40,000 require post burn care and hospitalization [1]. Depending on the severity of the burn, injury pathophysiology ranges from local tissue damage to a complex systemic response which can quickly become life threatening. Partial thickness and full thickness burn wounds (PTBWs and FTBWs, respectively) are frequently infected, significantly lengthening the healing time and often producing scars that can interfere with patient mobility, lead to long hospitalizations, and possibly requiring multiple reconstructive procedures [2]. Inflammation and infection generate free radicals and oxidative stress that slow granulation, tissue remodeling, and encourage scar formation. Prolonged healing time increases the risk of burn infections, contributing to patient morbidity and mortality [3].

Gram-positive and Gram-negative bacterial infections remain the most common causes of mortality following significant burn injury [4]. Wound biofilms interfere with wound healing, provide a barrier to antibiotic treatment, and if not managed effectively, may progress to chronic infection [5, 6]. Methicillin-resistant *Staphylococcus aureus* (MRSA) is the most common pathogen infecting PTBWs and FTBWs in the US [7]. Antibiotic resistance complicates MRSA treatment and allowing for formation of recalcitrant biofilms within the wound bed [8]. Current treatment options include debridement and curettage. Both approaches have limitations, as biofilms can recur within 24 hours of debridement, and curettage is a physically limited process [9]. Over the past decade, a variety of advanced burn dressings have been introduced, many of them containing silver compounds for their antimicrobial effects. While silver containing dressings have traditionally been effective in decreasing infection incidence, recent systematic review(s) indicate that silver-containing foam dressings and traditional silver sulfadiazine (SSD) dressings show no significant improvement in infected burn wound healing [10]. SSD, the standard of care for burn infections, has been associated with pain, burning sensation, itching, rash, and cytotoxicity [11]. Other topical antibiotic agents and therapeutic occlusive or exposure dressings are commonly used to facilitate healing and to prevent scar formation, although these therapies have resulted in variable clinical outcomes [12]. These varied observations make selection of the most appropriate dressing product a challenge for clinicians [13]. Ideally, a standard of care treatment for PTBWs should have the ability to maintain moisture balance, reduce inflammation, disrupt biofilm, and promote tissue re-epithelialization.

Traditional PTBW animal models used to investigate the therapeutic potential of these treatments have limitations [14, 15]. While porcine models are best representative of human PTBWs, the experiments have a high cost per animal as the number of wound sites per animal are limited by cross-contamination of wound sites [16]. While *ex vivo* models can be used to aid screening of different therapeutics and bacterial isolates of burn wound infections at early stages [14, 15], they lack robust live tissue to reflect the effect on wound healing. Although animal models do a poor job of replicating human scar formation, the pig model remains the most reliable to replicate the human response to therapeutics to study infection, inflammation and healing in burns [17]. Development of a porcine model that optimizes the number of wound sites per animal while minimizing wound cross-contamination would reduce cost and use of animals.

Chitosan, a glucosamine polymer derived from chitin (poly-N-acetylglucosamine), is an abundant, natural polymer that represents a promising source for the development of therapeutic wound agents as it is biodegradable and non-toxic [18]. Inherent to its chemical structure, chitosan has several basic amine groups giving it an overall cationic charge when in

acidic pH. Chitosan's cationic charge at low pH allows it to disrupt the outer membrane in Gram-negative and Gram-positive bacteria and increase bacterial cell membrane permeability [19], providing several beneficial anti-microbial effects. Chitosan's bactericidal effect against *Staphylococcus aureus* was shown to be a result of binding to the teichoic acids found in the bacterial cell wall [20, 21]. In addition to its bactericidal activity, chitosan has been observed to accelerate the wound healing process by stimulating macrophages and fibroblasts [22]. Further, the gene expression studies demonstrated improved healing through a process of reduced fibrosis and enhanced tissue regeneration. [23]. However, at physiological pH, chitosan has poor solubility and limited positive charge [18], thus limiting its application to acidic environments. To overcome these drawbacks, a water-soluble, polycationic chitosan derivative that maintains the positive charge well above a pH of 9 through derivatization with arginine has been developed and incorporated into a wound rinse and wound gel designed to treat traumatic and burn wounds. This arginine chitosan derivative is antimicrobial at physiologic pH [24] and has activity against a variety of biofilms [25–27].

The current study evaluates the efficacy of an arginine chitosan derivative wound rinse and wound gel designed to reduce bioburden and improve healing using a modified MRSA-infected porcine PTBW model. The results reported here offer progress towards a more efficient porcine wound model and a new therapeutic material to address the clinical challenges associated with treating MRSA-infected PTBW.

## Materials and methods

### Materials tested

FDA cleared 510(k) medical devices, SynePure™ Wound Cleanser (K143444) and Catasyn™ Advanced Technology Wound Hydrogel (K172338) (Synedgen, Claremont CA) containing a biocompatible chitosan derivatized with arginine as the proprietary ingredient, were used in the study. SynePure Wound Cleanser is an aqueous solution of chitosan-arginine with normal osmolality provided by a sorbitol, and with betaine as a biocompatible preservative. Catasyn Advanced Technology Wound Hydrogel is an aqueous gel comprised of hydroxypropyl methyl cellulose (hypromellose), preserved by methylparaben and betaine and brought to osmotic balance with sorbitol.

### Microtiter plate biofilm assay

The methicillin resistant *S. aureus* (MRSA) strain ATCC BAA-1717 was grown overnight in tryptic soy broth (TSB) and then adjusted to 1 McFarland turbidity standard, then further diluted 1:30 in TSB. The diluted culture was seeded into a 96-well microtiter plate with 3 wells for each treatment and controls. The biofilms were grown for 72 hours at 37°C incubator, without shaking. The 72-hr mature biofilms were washed twice gently with 200µl of 1X phosphate buffer saline (PBS) to remove any unbound bacteria. The biofilms were then treated with either 200µl of 1XPBS or the wound rinse and allowed to incubate for 1, 3, 5, or 10 minutes at room temperature. One set of replicates was left untreated in fresh TSB as a control. After each indicated treatment time, the wells were rinsed twice with 200µl of 1XPBS to remove unbound bacteria and dried for 2 hours at 37°C. Once dry, the biofilms were stained with 200µl 0.6% crystal violet and incubated for 15 minutes [28]. Stained biofilms were washed twice with 200µl of 1XPBS to remove unincorporated stain. Following this rinse, crystal violet was eluted by addition of 200µl of absolute methanol. The plates were closed and incubated at room temperature for 5 minutes. Eluted crystal violet was transferred into a fresh microtiter plate and the optical density (OD) of each well was determined at 590nm. Three independent experiments were performed, each in triplicate.

## Minimum Biofilm Eradication Concentration (MBEC)

MRSA (ATCC® BAA-1717™) was grown overnight in TSB and then adjusted to 1 McFarland turbidity standard, which was further diluted 1:30 in TSB. The diluted bacteria were added to the wells of a 96-well plate. Then the peg-lids of the MBEC plates were placed on top of the 96-well plate. The bacteria were allowed to grow on pegs for 72 hours at 37°C on a rocking table set to 3–5 rocks per minute with a 10° inclination. After the 72-hour incubation the peg lids were removed from the 96-well plate and placed into a fresh 96-well flat bottom microtiter plate containing 200µl of 1XPBS. Post rinsing, the peg lids were transferred onto 96 well plates containing the wound rinse, wound gel, or media control, and the plates were incubated at room temperature for 1, 3, 5 or 10, 30 and 60-minutes and 24 hours respectively. Following treatment, the peg lids were placed in a recovery plate containing 200µl of 1XPBS and soni-cated for 10 minutes to dissociate the remaining bacteria from the pegs. The peg lids were removed and kept aside. Sample aliquots from each well of the 96 well plates were serially diluted in sterile water and spot plated onto TSA plates. The plates were incubated for 24 hours at 37°C. The initial bacterial inoculum and biofilm controls were similarly quantified by serial dilution and spot plating. Three independent experiments were performed using three independent culture isolates.

## Animal study

**Animals.** Two female Yorkshire pigs (aged 6 months and weighing 60 kg) were used for the study in accordance with University of Pittsburgh, Division of Laboratory Animal Resources, IACUC standards (protocol #17030117). The animals were kept in smooth-sided, stainless-steel cages and fed a standard porcine diet. They were housed at a temperature of 20°C to 30°C with 65 percent humidity and a light cycle of 12hrs on/ 12hrs off.

**Generation and infection of partial-thickness burns.** The pigs were sedated with Keta-mine/Xylazine (20 mg/kg; 2 mg/kg) before being transferred onto an operating table. They received general anesthesia with 1.5–2.5% isofluorane in conjunction with oxygen. Intraoperative hydration was maintained with intravenous administration of Lactated Ringers solution at a flow rate of 5cc/kg/hr. Body temperature was maintained with the use of water blankets and bair hugger. Pig hair was shaved, and the pigs were washed with soap and water. Before wounding, the skin was surgically prepared using 7.5% povidone-iodine and 70% isopropyl alcohol for 3 minutes each. Using a prefabricated transparent sheet, the location of the wound was marked on the dorsum and flanks of the pig. An electric branding iron (40 x 40 x 5 mm) heated at ~ 200°C was applied for 10 seconds with a force of 1kg to produce consistent PTBWs. Each pig had 15 PTBWs. Immediately after burning, the loosely attached epidermis was removed by debridement, and the wounds were covered with gauze and wound chambers (S1 Fig).

## Wound chamber design and use

Wound chambers were engineered as a two-piece structure (base and cover) to surround, iso-late, and cover individual porcine wounds (S1 Fig). The wound chamber base was made of a stoma click barrier secured using tissue adhesive, moldable ring seal and staples. The wound chamber cover was made of gauze, transparent film dressing over the base's ring and secured with a stoma pouch. The pigs were then covered using cotton burn dressing (Medline, North-field, IL) secured with neck/forelimb elastic slings and goat jacket. On day 2 post-burning the pig wounds were bluntly debrided under sedation, cleaned with PBS, and inoculated with MRSA, on day 2 post- burn (S1 Fig).

## Inoculation of MRSA

Methicillin resistant *S. aureus* strain ATCC® BAA-1717™ (ATCC, Manassas, VA) working-stock was produced by culturing the bacteria on 0.5% sheep blood agar. A single colony was picked and propagated in trypticase soy broth. Bacterial viability was determined by assaying colony forming units (CFU) per milliliter of culture. Bacteria were cultured overnight at 37˚C in trypticase soy broth. The culture was centrifuged at 3000g for 5 min and pellet was resuspended in phosphate buffer saline (PBS). The bacterial suspension was diluted in PBS to obtain an OD600 range from 0.2 to 0.6 with an increment factor of 0.05. Serial dilution from each OD600 value were plated on tryptic Soy agar (TSA) plate and CFU was calculated following overnight incubation of plates at 37˚C. On the day of inoculation bacterial pellets were resuspended to obtain a bacterial concentration of $10^8$ CFU per ml in PBS. Three (3) ml of this bacterial suspension were then inoculated in each wound enclosed by chambers designed to retain the bacterial suspension and avoid cross contamination. Infection was allowed to develop in the wounds for 3 days.

## Experimental groups

Two pigs were used for all of the study groups, and all of the wounds were infected with MRSA. Starting post-MRSA inoculation day 3, each wound was assigned to one of three treatment groups (n = 10 each): 1) no treatment, 2) SSD, or 3) the combination of Synepure wound cleanser (wound rinse) and Catasyn advanced wound hydrogel (wound gel) (S2 Fig). Gauze was used as the dressing, and dressing changes were performed three times a week, with or without treatment agents.

**Quantitative assessment of MRSA load in tissue.**   Punch biopsies (3 mm in diameter) were collected from each wound upon each dressing change. Biopsies were weighed and added to 1ml PBS. Tissue was homogenized and serially diluted in PBS. 100 μl from each dilution was plated on sheep blood agar plates. Colonies were counted after overnight incubation at 37˚C. The number of colonies forming unit (CFU) per gram of tissue was calculated. Setting the initial bacterial CFU count before the start of treatment at day 3 as 100%, an improvement or decrease in the number of CFU with time was plotted.

**Wound assessment.**   Wounds were assessed for re-epithelialization and wound closure. Re-epithelialization was defined as macroscopic presence of new epithelium. Open wounds were defined as areas with unhealed wounds and no macroscopic evidence of epithelium or contraction. Tracings of designated wounds were taken at each dressing change on a transparent plastic sheet. Open wounds' surface area was quantified using ImageJ. Wound exudates and presence of pseudo eschar was qualitatively recorded. Wound progression was monitored clinically and was recorded with digital photographs. Images from day of sacrifice were compared to assess quality of wounds.

**Histology.**   Biopsy specimens from day 28 were fixed in 10% formalin followed in paraffin blocks, and then processed for standard Masson's trichrome staining and evaluated by light microscopy.

## Statistical analysis

All data were analyzed using IBM SPSS for Mac Version 25.0 (IBM Corp., Armonk, N.Y.). Descriptive statistics were used to compare wound size between groups. These were recorded as percentages for categorical variables and means and standard deviation for numerical variables. Categorical data was analyzed using Chi-square and Fisher's exact tests. Means of wound size were compared using ANOVA. Statistical significance was assumed for p-values < 0.05.

## Results

### The wound rinse reduces MRSA biofilm biomass *in vitro*

The wound rinse's activity against mature MRSA biofilms was investigated in an *in vitro* microtiter plate film bioassay. Treatment with the wound rinse resulted in a time-dependent exponential decrease in mature MRSA biofilm biomass compared to TSB and 1XPBS treated controls (Fig 1). A significant decrease in the MRSA biofilm biomass (53%) was observed within a minute of treatment with the wound rinse (P<0.0001) compared to the untreated control. Exposure of the biofilm to the wound rinse for 3- and 5- minutes resulted in 59% and 74% decreases, respectively, of the MRSA biofilm biomass (P<0.0001) compared to the untreated control. Application of the wound rinse for 10 minutes resulted in 81% reduction of the MRSA biofilm biomass compared to the untreated control (P<0.0001). No statistically significant differences were observed in the untreated controls over time.

### Wound rinse and wound gel disrupt mature *S. aureus* biofilm (MBEC)

A standard MBEC assay was used to further assess the effect of the wound rinse or wound gel on the viability of mature MRSA biofilms [29]. Treatment with the wound rinse for 1 minute resulted in significant reduction (2.5 log) in viable bacteria associated with the biofilms (P<0.0001). Within 10 minutes of application of the wound rinse, a >3 log reduction in CFU/ml was observed (P<0.0001) (Fig 2A). Treatment with the wound gel for 10 minutes resulted in a >3 log reduction in the number of viable bacteria within the biofilm (Fig 2B) (P<0.0001). Treatment for an hour resulted in complete eradication of the viable bacteria within the

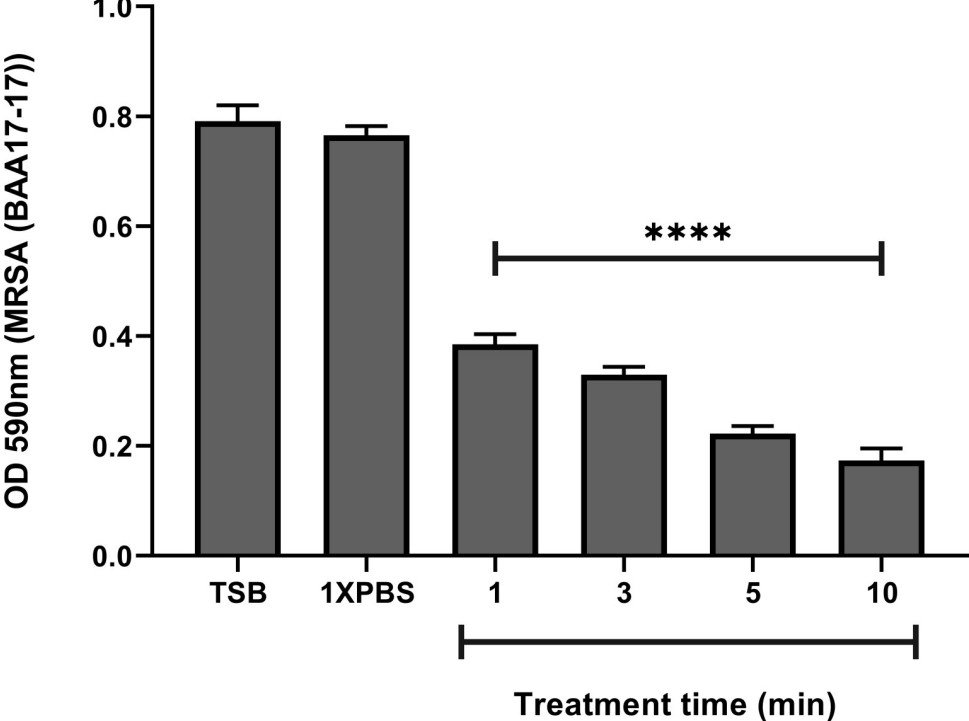

**Fig 1. Treatment with the wound rinse results in significant reduction of biofilm biomass of a 72-hour MRSA (ATCC® BAA-1717™) biofilm.** Significant reduction (P<0.0001) in MRSA biofilm biomass was observed at all time points tested compared to MRSA biofilms treated with 1XPBS or TSB. Data represented as the mean OD600 +/- SEM. **** $p < 0.0001$.

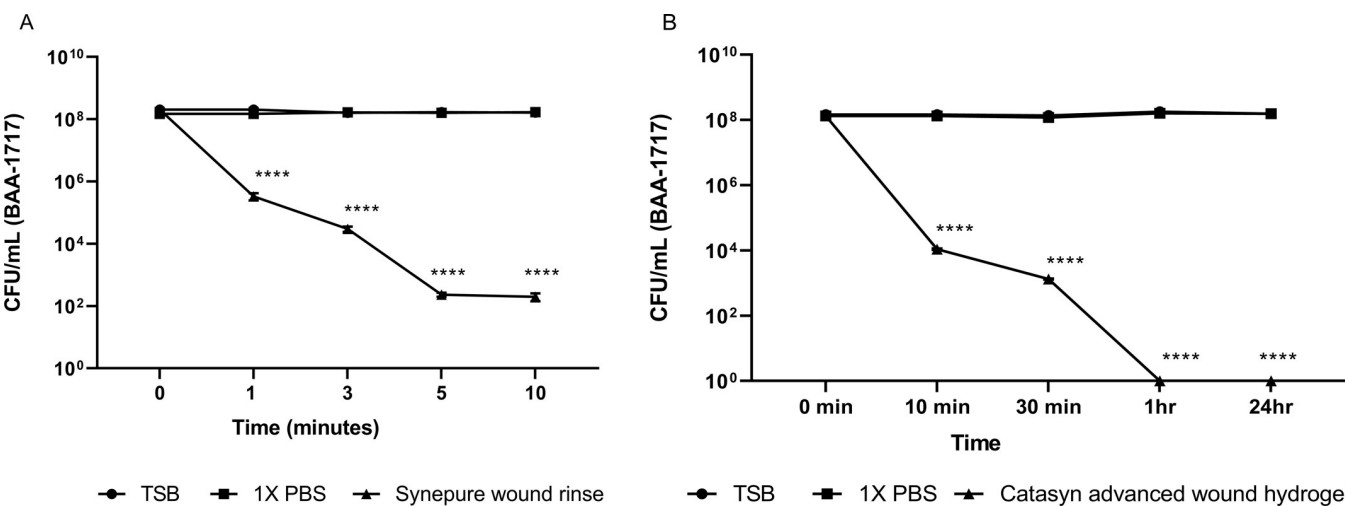

**Fig 2.** Treatment with wound rinse (A) and wound gel (B) lead to significant decrease in the viable bacteria in 72hr grown MRSA (ATCC® BAA-1717™) biofilm as reflected by remaining bacteria (CFU/ml). Data represented as the mean CFU/ml +/- SEM. ****, $p < 0.0001$.

MRSA biofilm (P<0.0001) (Fig 2B). Treatment with 1X PBS or TSB did not affect bacterial viability within the biofilm (Fig 2A and 2B). The difference in the reported treatment times in Fig 2A and 2B reflects the earliest point at which a statistically significant reduction of bacteria within the MRSA biofilms was observed. This point was achieved at 1 minute and 10 minutes when treated with wound rinse and wound gel, respectively.

## Treatment with the wound rinse and wound gel promotes PTBW closure *in vivo*

Treatment with the wound rinse/gel enhanced the rate of reduction in wound size leading to enhanced wound closure in MRSA-infected PTBWs compared to SSD treated and to untreated burn wounds (P<0.0001) (Fig 3). Treatment with wound rinse/gel resulted in significant decrease (P<0.0001) in wound size between day 7 and day 17 (P<0.0001) compared to the untreated and to SSD treated PTBW's (Fig 3). Infected burn wounds treated with the wound rinse/ gel showed faster healing rates, resulting in 67% reduction in wound size by day 11 and 89% reduction in wound size by day 17 as compared to SSD treated and untreated controls (Fig 3). The rate of wound closure in MRSA infected burn wounds treated with the wound rinse/gel combination was found to be ~7 days faster than similar wounds treated with SSD and 10 days faster than the untreated, MRSA infected burn wounds (Fig 3).

## Treatment with the wound rinse and wound gel reduces PTBW bacterial bioburden *in vivo*

Application of the wound rinse/gel resulted in significant decrease in the bacterial bioburden on the infected burn wound compared to wound infections treated with SSD (Fig 4). The rinse/gel combination resulted in a steady decrease, resulting in 3 log reductions in viable bacteria in infected burn wounds by day 11 which correlates with augmented wound healing compared to SSD treated infected burn wounds and untreated control. The combination rinse/gel treatment resulted in elimination of viable bacteria below the limit of detection from the infected burn wound within 21 days of treatment, compared to the untreated burn wounds and to the infected wound burns treated with SSD (24 days) (Fig 4).

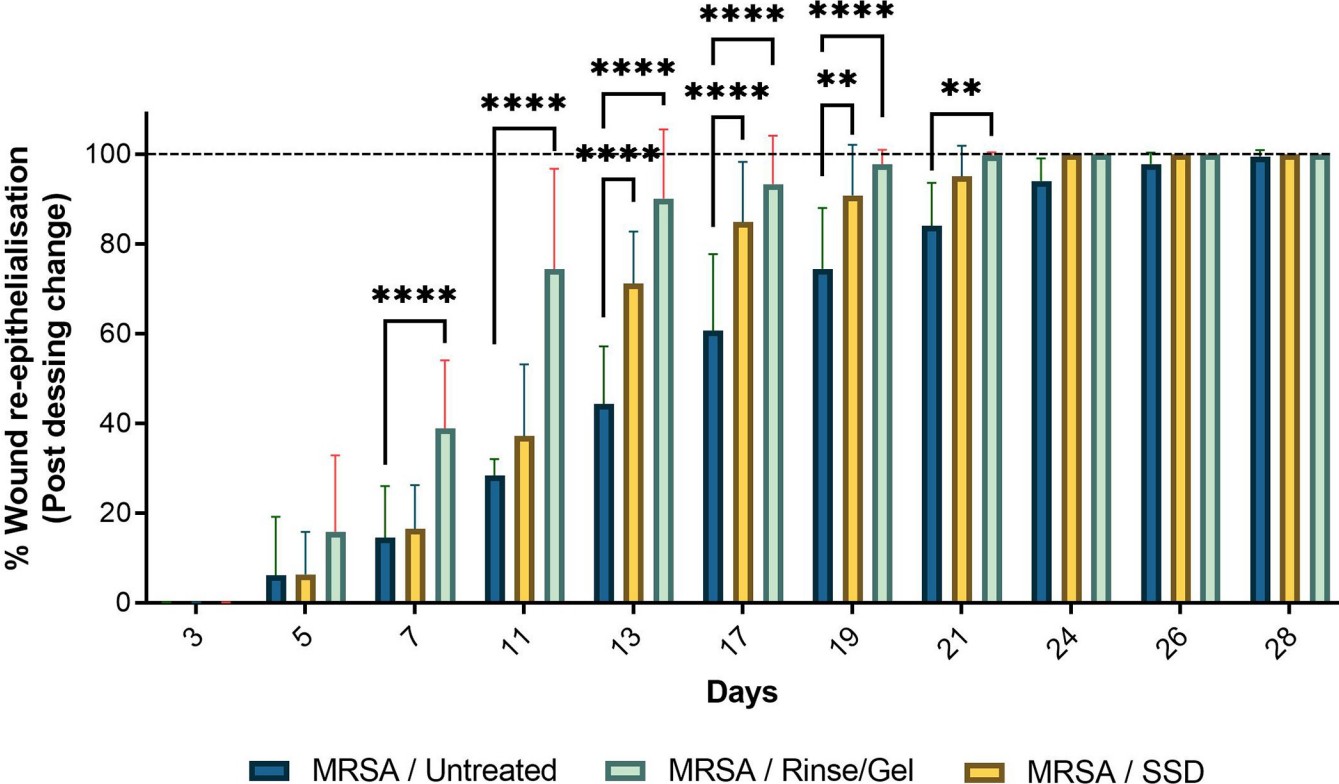

**Fig 3. Treatment with wound rinse and wound gel improved healing of MRSA-infected partial-thickness burn wounds.** Wound rinse/gel treatment accelerated the rate of wound closure compared to the silver sulfadiazine (SSD) and to untreated controls in MRSA-infected partial-thickness burn wounds. %, percentage; *, $p < 0.05$; **, $p < 0.01$; ***, $p < 0.001$; ****, $p < 0.0001$. Red (*)—statistical significance compared to SSD, and Black (*)- statistical significance compared to MRSA infected untreated. wounds.

## Treatment with the wound rinse and wound gel improves tissue re-epithelialization *in vivo*

Representative wound images in Fig 5A show the comparative effects of treatments over a period of 28 days. Although all treatment groups showed epithelialization by day 28, the final macroscopic quality of scars were inferior in the untreated and SSD groups compared to those treated with rinse/ gel. The rinse/ gel treatment promoted earlier healing resulting in a final scar with more normal pigmentation and vascularity (Fig 5A). Dressing changes demonstrated qualitatively cleaner wounds with less exudates in wounds treated with rinse/ gel compared to SSD or the untreated control (Fig 5A). PTBWs treated with the wound rinse/gel demonstrated no pseudo eschar formation and scarring when compared to wounds treated with SSD controls, making it easier to clean (Fig 5A).

PTBWs were also evaluated up to 28 days for complete re-epithelialization. The rate of re-epithelialization was observed to be significantly faster in wounds treated with the rinse/gel compared to SSD or the untreated control (Fig 5B) (P<0.0001). A significant increase in wound re-epithelialization was observed by day 7 in PTBWs treated with the rinse/gel compared to the untreated and to the SSD treated burn wounds (P<0.0001) (Fig 5B). By day 13, the wound rinse/gel treatment exhibited >80% wound re-epithelialization when compared to the untreated and to SSD treated burn wounds (P<0.0001) (Fig 5B).

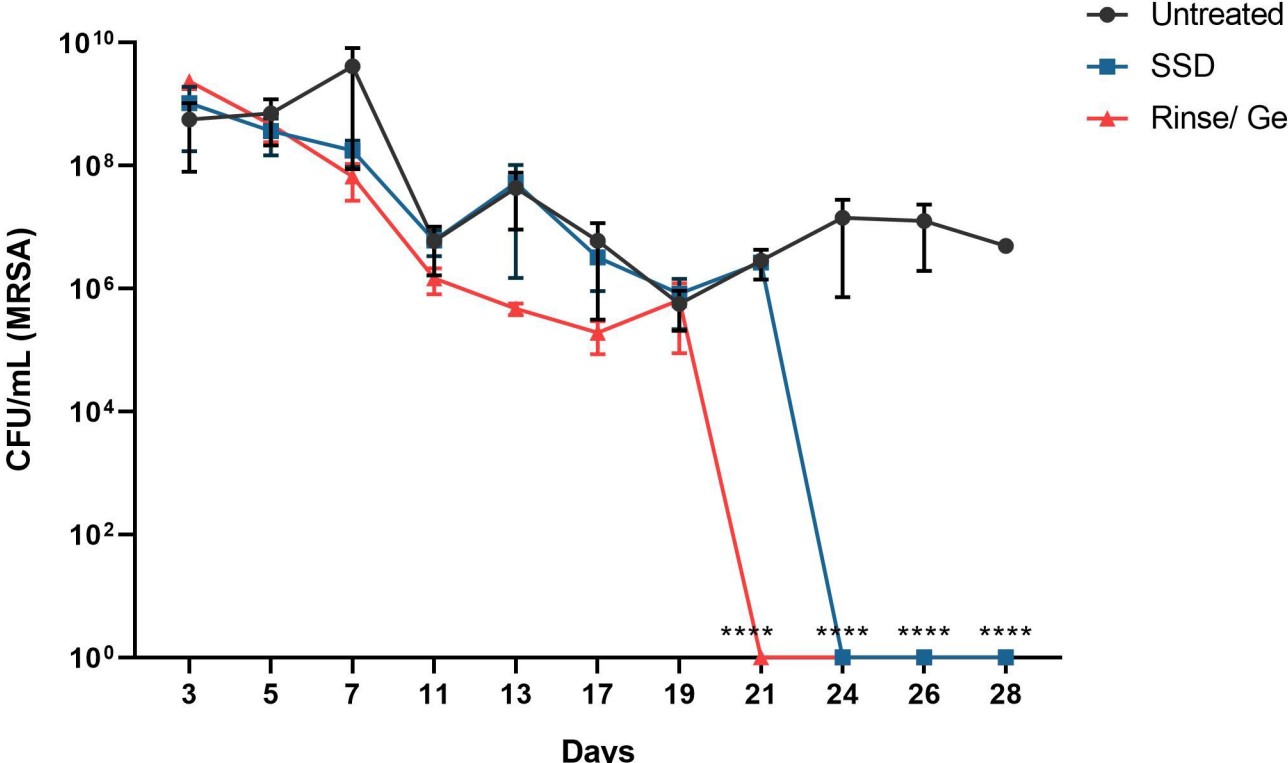

**Fig 4. The wound rinse and wound gel treatment resulted in significant reduction of bioburden in MRSA infected PTBWs.** Treatment with the wound rinse/ gel led to complete eradication of viable bacteria in PTBWs within 21 days of treatment compared to silver sulfadiazine (SSD) and untreated controls (P<0.0001). Data represented as the mean CFU/ml +/- SEM. ****, P<0.0001.

## PTBWs treated with the wound rinse and wound gel demonstrate better skin tissue organization and re-epithelialisation

On day 28, Masson's trichrome staining of the PTBWs was repeated. Relative to controls, the burns treated with the wound rinse/gel combination exhibited even thickness, epithelial evenness, and decreased epithelial density, positing a possible reduction in hypertrophic scar formation. In addition, histological staining demonstrates that the infected burns that received intervention exhibit improved tissue remodeling, clean dermo-epidermal margins, as well as a parallel and loose arrangement of collagen fibers in the dermis (Fig 6). This improved collagen content and organized architecture of collagen fibrils supports the observations of improved pseudo-eschar formation and improved healing of the interventional burns.

## Discussion

Burn injury is one of the most severe forms of dermal trauma and is often associated with significant pain and limitations in function [30]. Consequently, the investigation of advanced treatment options to evaluate their antibacterial and antibiofilm properties in addition to their wound healing properties is important. Traditional, non-porcine PTBW animal models used to investigate novel therapeutic options for burn infections are limited by poor replication of the human dermis burn environment and poorly replicate the human immune and healing response over time [31]. While porcine models are accepted as representative of human PTBWs, such experiments are often hindered by re-infection and cross-contamination of wound sites as well as differences in healing rates between animals. The *in vivo* wound

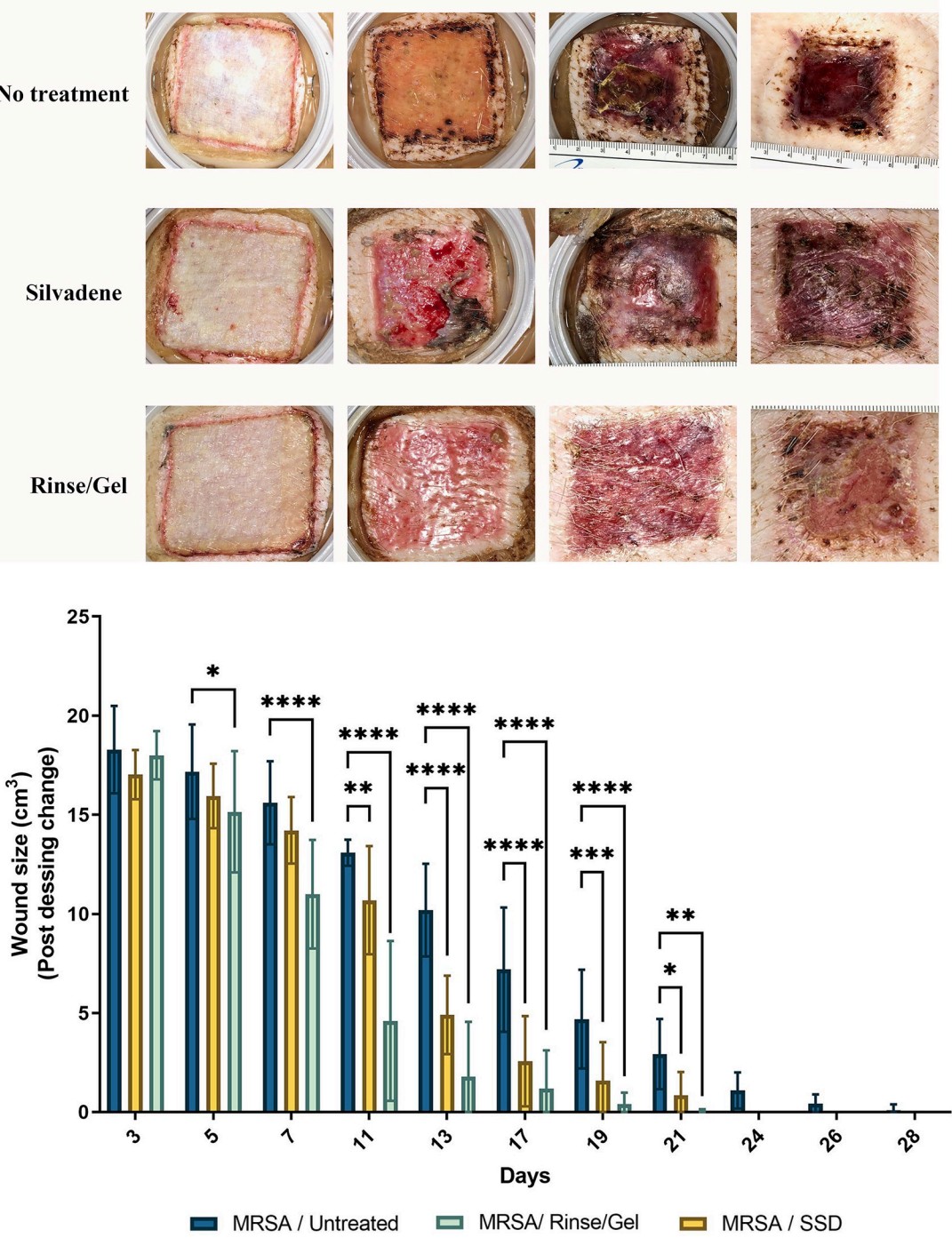

**Fig 5. A) Representative images of MRSA-infected PTBW's treated with SSD and wound rinse and wound gel compared to the untreated burns over 28 days, post dressing change.** Wound healing progression was imaged at the time of the post-dressing changes, following the initial injury. The rinse/gel treatment facilitates complete tissue re-epithelialization in infected PTBW by 28 days compared to silver sulfadiazine (SSD) treated wounds that show healthy granulation, but only a thin layer of new epithelium. POD- Post dressing change. Wound dimensions- 4X4 cm. **B) Rate of wound re-epithelialization post dressing change over 0–28-day pigs.** Rinse/gel treatment consistently resulted in faster tissue re-epithelialization over time (****, P<0.0001; ***, P< 0·001) compared to the SSD treated and untreated controls, respectively.

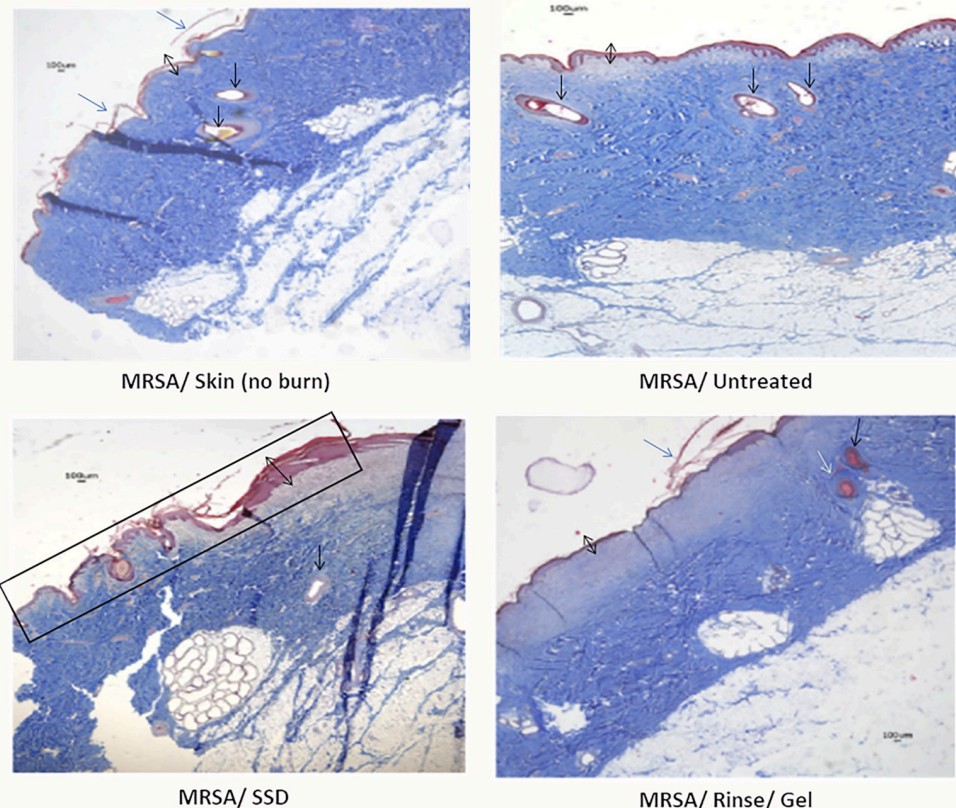

**Fig 6. Comparative histological images reveal the difference in healing, between MRSA-infected PTBWs treated with wound rinse and wound gel or with silver sulfadiazine (SSD) compared to the untreated wound burns.** Skin biopsies of burn wounds from day 28 were subjected to the Masson's trichrome staining. Masson trichrome staining highlights collagen content (blue). The black double headed arrow represents the thickness of epidermis, the single headed blue arrow shows the keratinized zone of skin, the single headed white arrow shows follicles, the single headed black arrow projects the sebaceous gland, and the black box shows the damage to collagen by the burn that remains unhealed in SSD treated samples. The pictures show thicker collagen fiber pattern, even epithelial thickness, and decreased epithelial cell density (MRSA/Rinse/Gel) in the intervention burn, compared to the control images.

chamber animal model outlined in the study (S1 Fig) was shown to be an effective tool to simplify assessment of the therapeutic potential of multiple compounds and bacterial isolates in PTBW infections. The model was designed to reduce cost while still using a representative porcine model and to be accessible in a variety of research environments. The PTBW porcine model described here isolates multiple sites independently and allows for a higher density of wounds on a single animal. The chamber model is engineered to accommodate numerous large and deep wounds, infected with potentially multiple different strains of bacteria, and support multiple therapeutic treatments to be studied without cross contamination. In addition to the armament of tools to study wound healing, infection, and inflammation, this model potentially reduces research cost by: (a) reducing the number of animals required to attain a significant number of PTBWs, (b) reducing the number of animals required to test multiple agents with less risk of wound cross-contamination, and (c) limiting the amount of time spent by researchers and veterinarian technicians during dressing changes. Moreover, limiting cross-contamination reduces potential resources wasted on non-viable PTBWs.

Many chitosan-based wound care solutions have been proposed but the requirement of acidic pH for activity limits chtiosan's applicability [32]. This study assessed a chitosan derivative with stable, polycationic charge at a wide range of pH and with excellent solubility to

address these limitations. Further, the wound rinse and wound gel used in this study are already FDA cleared wound care products. (SynePure Wound Cleanser (K143444) and Catasyn Advanced Technology Wound Hydrogel (K172338).

This study demonstrates *in vitro* and *in vivo* the ability of the wound rinse and the wound gel independently and in combination to significantly reduce the viable bacteria present in the bacterial biofilms as well as to reduce biomass. Exposure of 72 hour grown MRSA biofilms to the rinse *in vitro* resulted in significant ($P<0.0001$) decrease in biomass compared to exposure to 1XPBS or untreated control within 1 min of treatment (Fig 1). Further assessment of the rinse also showed significant reduction in viable bacteria within the biofilms within 5 minutes of treatment time (Fig 2A). Exposure to the gel *in vitro*, resulted in complete eradication of the viable bacteria in the MRSA biofilm within 1hr of treatment time ($P<0.0001$) (Fig 2B). *In vivo* treatment of the MRSA infected PTBWs with wound rinse followed by the gel resulted in significant reduction of bacterial burden by day 7, and complete eradication of the viable bacteria in 21 days of application, compared to SSD and untreated control (Fig 4).

The ability of the rinse/gel treatment to disrupt biofilms *in vitro* and to reduce MRSA infections in the porcine model is noteworthy for its potential use in overcoming the recalcitrant nature of biofilm-associated sepsis and resulting in delayed wound healing. As shown in Fig 4A and 4B, the chitosan rinse/gel combination was superior to SSD in its ability to kill or remove bacteria and to promote wound healing.

The *in vivo* studies highlight the impact of the wound rinse/ gel on wound healing (Fig 3). Significant reduction in viable bacteria (>3 log reduction) correlated with significantly improved wound healing by day 11 ($P<0.0001$) in MRSA infected porcine PTBWs treated with the rinse/gel treatment compared to SSD and to untreated controls (Fig 3). Furthermore, the rinse/gel treatment facilitated complete wound healing by day 17 ($P<0.0001$) compared to SSD and to untreated controls (Fig 3). On analyzing the representative images of the post-operative (Day 28) infected wounds, those treated with the rinse/gel treatment demonstrated a stronger resemblance to the dermal and epidermal structures of native skin (Fig 5A).

Re-epithelialization of a burn wound is a critical step in the wound healing process in order to restore native skin structure. The wound rinse/gel treated PTBWs show accelerated tissue re-epithelialization ($P<0.0001$) by day 7 when compared to SSD treated or to untreated PTBW's (Fig 5B). The MRSA infected, untreated wound showed slower healing as reflected by delayed reduction of wound size (Fig 3) and slower progression in wound re- epithelialization (Fig 5B). The persistence of MRSA infection in these SSD or untreated wounds likely contributed to the delayed wound tissue re-epithelialization (Fig 4).

Though Figs 3 and 5B show a 100% reduction in wound size and a 100% increase in re- epithelialization by day 28 in all wounds, the macroscopic quality of scars was inferior in the untreated and SSD groups compared to those treated with the wound rinse/gel. The final scar in burn wounds treated with the wound rinse/gel had normal pigmentation and vascularity, lower exudates, and no signs of pseudo eschar formation or scarring as compared to untreated wounds or wounds treated with SSD (Fig 5A). Histological analyses of the burn wounds on day 28 showed that burn wounds treated with the combination rinse/gel treatment exhibited significantly poor tissue remodeling to the epidermis and dermis layers and enhanced healing compared to the wounds treated with SSD and the untreated controls (Fig 6).

MRSA biofilms are often associated with impaired epithelialization and granulation tissue formation, leading to a low-grade inflammatory response that interferes with wound healing. The effect of the combination rinse/gel treatment starting day 7 resulted in > 3 log reduction in viable bacteria in the infected wound that correlated with significant enhancement of wound healing and re-epithelialization observed on days 7 through 21. While frequent physical removal of wound biofilm and appropriate antibiotic and topical antimicrobial therapies

are best practice today, novel products are needed that can disrupt complex biofilm communities, avoid generation of antimicrobial resistance, and prevent re-infection in non-healing wounds [33–35]. With its ability to facilitate healing, to significantly reduce biomass and viable bacteria in the biofilm and to reduce bacterial burden (below the limit of detection) in infected burn wounds, the wound rinse and gel combination has proven to be a promising therapeutic option for treating PTBW infections.

## Supporting information

**S1 Fig. Wound chambers design and use.** Wound chambers were engineered as a two-piece structure (base and cover) to surround, isolate, and cover individual porcine wounds.
(TIF)

**S2 Fig. Experimental treatment groups.**
(TIF)

**S3 Fig.**
(TIF)

**S1 File.**
(PDF)

## Acknowledgments

The University of Pittsburgh thanks the Army, Navy, NIH, Air Force, VA and Health Affairs for their support.

## Author Contributions

**Conceptualization:** Francesco Egro, Vidya Narayanaswamy, Stacy M. Towsend, Shenda Baker, Peter Rubin.

**Data curation:** Francesco Egro, Alex Repko, Vidya Narayanaswamy, Asim Ejaz, Deokyeol Kim, M. Asher Schusterman, Ali Ayyash, Shenda Baker, Jenny Ziembicki, Kacey Marra, Peter Rubin.

**Formal analysis:** Francesco Egro, Alex Repko, Vidya Narayanaswamy, Asim Ejaz, Shenda Baker, Jenny Ziembicki, Peter Rubin.

**Funding acquisition:** Peter Rubin.

**Investigation:** Francesco Egro, Alex Repko, Vidya Narayanaswamy, Asim Ejaz, Deokyeol Kim, M. Asher Schusterman, Ali Ayyash, Stacy M. Towsend, Shenda Baker, Jenny Ziembicki, Kacey Marra, Peter Rubin.

**Methodology:** Francesco Egro, Alex Repko, Vidya Narayanaswamy, Asim Ejaz, Deokyeol Kim, M. Asher Schusterman, Ali Ayyash, Stacy M. Towsend, Shenda Baker, Jenny Ziembicki, Kacey Marra, Peter Rubin.

**Project administration:** Francesco Egro, Vidya Narayanaswamy, Shenda Baker, Peter Rubin.

**Resources:** Francesco Egro, Vidya Narayanaswamy, Shenda Baker, Kacey Marra.

**Software:** Francesco Egro, Shenda Baker.

**Supervision:** Francesco Egro, Stacy M. Towsend, Shenda Baker, Kacey Marra, Peter Rubin.

**Validation:** Francesco Egro, Vidya Narayanaswamy, Stacy M. Towsend, Shenda Baker, Jenny Ziembicki, Kacey Marra.

**Visualization:** Vidya Narayanaswamy, Stacy M. Towsend, Shenda Baker, Jenny Ziembicki, Kacey Marra, Peter Rubin.

**Writing – original draft:** Vidya Narayanaswamy, Shenda Baker.

**Writing – review & editing:** Francesco Egro, Vidya Narayanaswamy, Allister Loughran, Shenda Baker, Kacey Marra, Peter Rubin.

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
