## [Decision Letter · Decision Letter 0]

25 May 2022

PONE-D-22-02846Soluble chitosan derivative prevents wound infections and promotes wound healing in a novel MRSA- infected porcine partial- thickness burn wound model.PLOS ONE

Dear Dr. Rubin, Thank you for submitting your manuscript to PLOS ONE. After careful consideration, we feel that it has merit but does not fully meet PLOS ONE’s publication criteria as it currently stands. Therefore, we invite you to submit a revised version of the manuscript that addresses the points raised during the review process.

Specifically, please include more information on the used product. The manuscript should be revised to include the recent relevant work and to discuss the added value of the current work. All applied standardized methods should be cited. All figures should be clearly displayed, and all relevant figures should be included within the manuscript. The committee approval for the animal study should be included.

We look forward to receiving your revised manuscript.

Kind regards,

Amal Al-Bakri

Academic Editor

PLOS ONE

**Journal requirements:**

“The work was funded by the United States Department of Defense, through the Armed Forces Institute of Regenerative Medicine (AFIRM), under Award No. W81XWH-14-2-0004.”

Please state what role the funders took in the study.  If the funders had no role, please state: ""The funders had no role in study design, data collection and analysis, decision to publish, or preparation of the manuscript.

“Source of funding: The work was funded by the United States Department of Defense, through the Armed Forces Institute of Regenerative Medicine (AFIRM), under Award No. W81XWH-14-2-0004.”

“The work was funded by the United States Department of Defense, through the Armed Forces Institute of Regenerative Medicine (AFIRM), under Award No. W81XWH-14-2-0004.”

“VPN and SMB are paid employees of Synedgen. SMB have ownership and patents affiliated with Synedgen and is also a board member. The potential conflicts noted have not impacted or influenced the findings of this manuscript. For the remaining authors none are declared”

5. PLOS requires an ORCID iD for the corresponding author in Editorial Manager on papers submitted after December 6th, 2016. Please ensure that you have an ORCID iD and that it is validated in Editorial Manager. To do this, go to ‘Update my Information’ (in the upper left-hand corner of the main menu), and click on the Fetch/Validate link next to the ORCID field. This will take you to the ORCID site and allow you to create a new iD or authenticate a pre-existing iD in Editorial Manager. Please see the following video for instructions on linking an ORCID iD to your Editorial Manager account: https://www.youtube.com/watch?v=_xcclfuvtxQ.

Reviewers' comments:

Reviewer's Responses to Questions

**Comments to the Author**

1. Is the manuscript technically sound, and do the data support the conclusions?

Reviewer #1: Yes

Reviewer #2: No

2. Has the statistical analysis been performed appropriately and rigorously? 

Reviewer #1: Yes

Reviewer #2: Yes

3. Have the authors made all data underlying the findings in their manuscript fully available?

Reviewer #1: Yes

Reviewer #2: Yes

4. Is the manuscript presented in an intelligible fashion and written in standard English?

Reviewer #1: Yes

Reviewer #2: No

5. Review Comments to the Author

Reviewer #1: The paper by Francesco M. Egro et al descibes the effect of soluble chitosan derivative on wound infections and wound healing in a burn wound model. Overall the experimental design is relevant and adeqatelly described, and idea is publication-worth.

There are some questions which should be rized before the consideration for acceptance.

The main question is regarding the characteristic of Wound Cleanser and Catasyn Advanced Technology Wound Hydrogel. Although the reviewer realize that the exact formulation can be closed by patent, some general information should be provided, like the main active compound, in what solution, the molecular weight of chitosan etc. Without this information the results cannot be proven and scientific value of the work is low.

The introduction should be improved by addition of references to works showing the positive effect of chitosan on infected wounds healing, and anti-biofilm activity. A simple search in google scholar gave many works close to this paper. Please site them, may be, excluding those where silver was used as its negative effect is postulated in introduction: https://scholar.google.com/scholar?q=chitosan+antibiofilm+wound+healing&hl=en&as_sdt=0

line 120 add reference to CV-staining protocol

Materials and methods - please give references to similar works if possible, especially in experiments with animals.

244-250. Please add a short explanation why different treatment time was chosen for the rinse and gel.

Fig 3 - It would be worth to add a panel with images of wounds. If these data were obtained form those images which are shown on Fig 5 - please put them together

Reviewer #2: Dear authors,

Thank you for submitting your manuscript. However, I am returning it to you without further review in order not to delay its eventual publication. Therefore, I do not think that the manuscript is suitable to the journal and rejected due to below-following points:

1. The abstract should be rewritten to focus on the aim of this study. This study does not show the main problem to be solved.

2. The novelty of this study must be highlighted. MRSA-infected porcine PTBW model and materials prepared from chitosan derivative were researched popularly and applied for many medicines applications.

3. In the line 341 and 342, the authors said, “the model was designed to reduce cost”. authors should explain more clearly. How does the model reduce cost from this study?

4. Line 347 and 348, the authors said, “this model can be used to save time and reduce animal resources”. Another animal resources will be chosen for different purposes.

5. The figures in the manuscript are not clear and the quality of images should be improved. Furthermore, figure S1 is not clear to read.

6. Language needs substantial improvement.

7. There are many spelling mistakes in the manuscript. Authors should replace “was” word by “were” word in line 119; “scars were inferior” by “scars was inferior” line 291, “treatment lead to” by “treatment led to” in line 303; “dressing change” by “dressing changes” in line 312; “bacteria present in” by “bacteria presents in” in line 354.

Thank you, once again, for submitting the manuscript.

Dr. Thi-Hiep NGUYEN

6. PLOS authors have the option to publish the peer review history of their article (what does this mean?). If published, this will include your full peer review and any attached files.

Reviewer #1: No

Reviewer #2: No

---

## [Author Response · Author response to Decision Letter 0]

3 Aug 2022

Dear Editor, 

Thank you for your supportive comments and those of the reviewers. We are grateful for the kind and helpful review that supported the clarification of specific points in our manuscript. We have carefully considered the reviewer’s comments and have addressed each one specified below.

Reviewer #1

1. The main question is regarding the characteristic of Wound Cleanser and Catasyn Advanced Technology Wound Hydrogel. Although the reviewer realize that the exact formulation can be closed by patent, some general information should be provided, like the main active compound, in what solution, the molecular weight of chitosan etc. Without this information the results cannot be proven and scientific value of the work is low.

Lines118-125- Additional information on Wound Cleanser and Catasyn Advanced Technology Wound Hydrogel have been included.

2. The introduction should be improved by addition of references to works showing the positive effect of chitosan on infected wounds healing, and anti-biofilm activity. A simple search in google scholar gave many works close to this paper. Please site them, may be, excluding those where silver was used as its negative effect is postulated in introduction: https://scholar.google.com/scholar?q=chitosan+antibiofilm+wound+healing&hl=en&as_sdt=0

Introduction has been revised and additional references have been included throughout the paper- Lines 98, 102, 107, 108, 138, 367, F372.

3. line 120 add reference to CV-staining protocol

Reference has been added.

4. Materials and methods - please give references to similar works, if possible, especially in experiments with animals.

Prior studies demonstrating the benefit of chitosan were discussed in the introduction. “Fu et al showed that chitosan polymers have bactericidal effect against Staphylococcus aureus by binding to the teichoic acids found in the bacterial cell wall (20, 21). In addition to the bactericidal activity chitosan has been observed to accelerate the wound healing process by stimulating inflammatory cells, macrophages, and fibroblasts, hence boosting the inflammatory phase (22). However, at physiological pH, the applications of chitosan are limited due to its poor solubility and limited positive charge (18).”

Our study is the first to our knowledge assessing the impact of chitosan on burn wound healing. The following has been added to the discussion on line 359-360 “Furthermore, no study to our knowledge has determined the impact of chitosan on burn wound healing”.

5. 244-250. Please add a short explanation why different treatment time was chosen for the rinse and gel.

The difference in the treatment times in Figure 2A and 2B reflects the earliest point at which there was a statistically significant reduction of bacteria within the MRSA biofilms. This point was achieved at 1 minute and 10 minutes when treated with wound rinse and wound gel, respectively

6. Fig 3 - It would be worth to add a panel with images of wounds. If these data were obtained from those images which are shown on Fig 5 - please put them together.

Reviewer #2

1. The abstract should be rewritten to focus on the aim of this study. This study does not show the main problem to be solved. 

Abstract has been revised focusing on the problem to be solved. 

2. The novelty of this study must be highlighted. MRSA-infected porcine PTBW model and materials prepared from chitosan derivative were researched popularly and applied for many medicines applications. 

The novelty of this study has been highlighted in lines 357-358, 362-364, 376-380.

3. In the line 341 and 342, the authors said, “the model was designed to reduce cost”. authors should explain more clearly. How does the model reduce cost from this study?

More clarity around this has been provided in lines 370-374.

4. Line 347 and 348, the authors said, “this model can be used to save time and reduce animal resources”. Another animal resources will be chosen for different purposes. 

More clarity around this has been provided in lines 370-374.

5. The figures in the manuscript are not clear and the quality of images should be improved. Furthermore, figure S1 is not clear to read.

We are including high quality images. S1 was removed due to low quality. S2 renamed to S1, S3 renamed to S2. The manuscript has been updated.

6. Language needs substantial improvement. 

Revisions have been made throughout the paper to improve the language.

7. There are many spelling mistakes in the manuscript. Authors should replace “was” word by “were” word in line 119; “scars were inferior” by “scars was inferior” line 291, “treatment lead to” by “treatment led to” in line 303; “dressing change” by “dressing changes” in line 312; “bacteria present in” by “bacteria presents in” in line 354.

Minor edits and grammatical revisions have been made throughout the paper.

Lines 1, 18, 26, 27, 31, 32,40, 48, 51, 55, 56, 59, 61, 89, 92, 94, 96- 98, 119, 147, 161, 181, 187, 207, 263, 291, 303, 308, 312, 323-326, 353, 358, 361, 371, 372, 376-378, 399-406, 408, 427- 428, 434-435, 440.

---

## [Decision Letter · Decision Letter 1]

30 Aug 2022

Soluble chitosan derivative treats wound infections and promotes wound healing in a novel MRSA- infected porcine partial- thickness burn wound model.

PONE-D-22-02846R1

Dear Dr. RUBIN,

We’re pleased to inform you that your manuscript has been judged scientifically suitable for publication and will be formally accepted for publication once it meets all outstanding technical requirements.

Kind regards,

Amal Al-Bakri

Academic Editor

PLOS ONE

Additional Editor Comments (optional):

Reviewers' comments:

Reviewer's Responses to Questions

**Comments to the Author**

1. If the authors have adequately addressed your comments raised in a previous round of review and you feel that this manuscript is now acceptable for publication, you may indicate that here to bypass the “Comments to the Author” section, enter your conflict of interest statement in the “Confidential to Editor” section, and submit your "Accept" recommendation.

Reviewer #1: All comments have been addressed

2. Is the manuscript technically sound, and do the data support the conclusions?

Reviewer #1: Yes

3. Has the statistical analysis been performed appropriately and rigorously? 

Reviewer #1: Yes

4. Have the authors made all data underlying the findings in their manuscript fully available?

Reviewer #1: Yes

5. Is the manuscript presented in an intelligible fashion and written in standard English?

Reviewer #1: Yes

6. Review Comments to the Author

Reviewer #1: (No Response)

7. PLOS authors have the option to publish the peer review history of their article (what does this mean?). If published, this will include your full peer review and any attached files.

Reviewer #1: No

---

## [Editor Report · Acceptance letter]

5 Oct 2022

PONE-D-22-02846R1 

Soluble chitosan derivative treats wound infections and promotes wound healing in a novel MRSA-infected porcine partial-thickness burn wound model. 

Dear Dr. Rubin:

I'm pleased to inform you that your manuscript has been deemed suitable for publication in PLOS ONE. Congratulations! Your manuscript is now with our production department. 

Kind regards, 

on behalf of

Dr. Amal Al-Bakri 

Academic Editor

PLOS ONE